# Alarm communication predates eusociality in termites

David Sillam-Dussès[1,10], Vojtěch Jandák[2,10], Petr Stiblik[3], Olivier Delattre[1], Thomas Chouvenc[4], Ondřej Balvín[5], Josef Cvačka [6], Delphine Soulet[1], Jiří Synek[3], Marek Brothánek [2], Ondřej Jiříček[2], Michael S. Engel [7✉], Thomas Bourguignon[8,9] & Jan Šobotník [8✉]

Termites (Blattodea: Isoptera) have evolved specialized defensive strategies for colony protection. Alarm communication enables workers to escape threats while soldiers are recruited to the source of disturbance. Here, we study the vibroacoustic and chemical alarm communication in the wood roach *Cryptocercus* and in 20 termite species including seven of the nine termite families, all life-types, and all feeding and nesting habits. Our multi-disciplinary approach shows that vibratory alarm signals represent an ethological synapomorphy of termites and *Cryptocercus*. In contrast, chemical alarms have evolved independently in several cockroach groups and at least twice in termites. Vibroacoustic alarm signaling patterns are the most complex in Neoisoptera, in which they are often combined with chemical signals. The alarm characters correlate to phylogenetic position, food type and hardness, foraging area size, and nesting habits. Overall, species of Neoisoptera have developed the most sophisticated communication system amongst termites, potentially contributing to their ecological success.

[1] University Sorbonne Paris Nord, Laboratory of Experimental and Comparative Ethology UR4443, 93430 Villetaneuse, France. [2] Czech Technical University in Prague, Faculty of Electrical Engineering, 166 27 Prague 6, Czech Republic. [3] Faculty of Forestry and Wood Sciences, Czech University of Life Sciences Prague, 165 21 Prague 6 - Suchdol, Czech Republic. [4] Entomology and Nematology Department, Fort Lauderdale Research and Education Center, University of Florida, Institute of Food and Agricultural Sciences, Fort Lauderdale, Florida 33314, USA. [5] Faculty of Environmental Sciences, Czech University of Life Sciences Prague, 165 21 Prague 6 - Suchdol, Czech Republic. [6] Institute of Organic Chemistry and Biochemistry of the Czech Academy of Sciences, 166 10 Prague, Czech Republic. [7] Division of Entomology, Natural History Museum, and Department of Ecology & Evolutionary Biology, 1501 Crestline Drive—Suite 140, University of Kansas, Lawrence, Kansas 66045, USA. [8] Faculty of Tropical AgriSciences, Czech University of Life Sciences Prague, 165 21 Prague 6 - Suchdol, Czech Republic. [9] Okinawa Institute of Science and Technology Graduate University, Okinawa, Japan. [10]These authors contributed equally: David Sillam-Dussès, Vojtěch Jandák. ✉email: msengel@ku.edu; sobotnik@ftz.czu.cz

To ensure communal living, the existence of a common defense among group members and against any threat is essential. Defensive strategies range from construction of protective barriers to coordinated responses to a particular threat, a predator, competitor, or pathogen. These responses require complex coordination, either in behavioral repertoires or physiological responses. In addition, they may be associated with concomitant morphological specializations which are obtained by building on existing or coopting new developmental pathways to achieve such functions. In essence, a successful defense, whether of a single mother defending her brood or a colony of millions, is the result of a plethora of evolutionary changes honed to increase the collective survivorship of the individuals participating in communal life[1,2].

In contrast to solitary animals, which can only rely upon themselves to protect against predators, some animals in social groups may dedicate themselves exclusively to foraging under the protection of specialized conspecifics. In the case of danger, the latter alert the former by alarm signaling. Such task partitioning allows the group to be most efficient at low risk[3]. This kind of communication amongst conspecifics is called alarm communication or alarm signaling. It is a defensive strategy that has evolved independently in many social animals, either vertebrates or arthropods, as it increases the fitness of social groups[4–12]. An alarm signal emitted by a colony member can make conspecifics aware of danger[13]. Thus, such a signal can be used by nearby conspecifics to rapidly respond to by displaying defensive or evading behaviors in order to prevent or limit casualties[14]. Usually, the first animal alarm signals that come to mind are the familiar alarm calls. These acoustic signals are common in mammals and birds[4,15], such as the classic example of the alarm calls in Vervet monkeys[16].

Alarm signals may also be transmitted in many animals by two distinct sensory channels: vibroacoustic and chemical[17,18]. Vibroacoustic communication involves substrate- and/or air-borne vibrations. It is considered as the most ancient and taxonomically widespread form of communication[19], and more than 90% of insects use substrate-borne vibrations alone or in concert with other forms of signaling[20]. In eusocial insects, the vibroacoustic signals act as either short-range (tactile) signals[21–23], or long-range vibrations perceived by distant nestmates through the Johnston's (air-borne) or subgenual (substrate-borne) organs[24,25]. Vibroacoustic signaling may carry various messages, such as alarm, recruitment, or begging for food[23,26–28]. Specific means of vibroacoustic communication were observed in termites but not in other social insects: alarming nestmates in response to pathogen encounter[29], evaluation of volume of the remaining wood[30] or perception of approaching competitor[31]. Apart of stridulatory organs in many ant groups, vibroacoustic signals are generated by inconspicuous body parts showing little to no specialization to this particular task[23].

Chemical alarm signals are widespread in animals, including insects[17,32,33]. The release of a volatile substance, an alarm pheromone, warns conspecifics of danger. Alarm pheromones provoke strong dose- and context-specific responses, resulting in retreat or attack, the latter usually accompanied with fast changes in caste or age-category proportions of the insects involved[34–37]. While the glandular origins of the alarm pheromones are diverse and taxon-specific[37–40], it is important to note that all alarm pheromones are produced by abdominal glands in cockroaches, while exclusively by cephalic glands in termite soldiers[27,34,41,42].

All social insects have evolved complex defensive traits, including morphological adaptations, chemical defenses, structural nest modifications, and alarm behaviors, as inherent components of colony defense[43–46]. Both vibroacoustic and chemical signals occur in all major eusocial insect groups (termites, ants, bees, and wasps). These signals are responsible for the alarm behavior, defined for termites by Deligne and coauthors[47] as a specific behavior implying the emission of a signal by an individual that has experienced a dangerous situation, the perception of this signal by distant nestmates, and subsequent adaptive modifications in their behavior. Such modifications consist of a general increase in the level of activity, locomotive changes, aggregation, flight, and recruitment of other termites to the site of the disturbance. Thus, the emergence of specialized defensive castes and complex behaviors[34,41,47] presumably contributed to the ecological success of termites and led to their extraordinary abundance throughout the tropics[11,12,47–50].

Unlike social Hymenoptera, termites are hemimetabolous insects and the foraging parties comprise juvenile individuals, which are wingless and largely unsclerotized[51]. The first line of termite defense is passive, and consists in the physical isolation of the colony from the hostile environment, built and maintained by vulnerable workers[34]. The active defense strategies are best exemplified by large soldiers (such as in *Mastotermes*, *Macrotermes*, *Syntermes*, *Cornitermes*, *Labiotermes*, or *Cubitermes*), which can inflict deep wounds with their mandibles, often coupled with the release of toxic or anti-healing compounds making termite bites truly unforgettable[34,52], many personal observations from D.S.D. and J.Š. These toxins may be released in copious amounts, as is the case in *Coptotermes*, in which the frontal gland secretion represents over a third of soldier fresh body weight[53]. The sticky toxic and irritating content of the frontal gland is sprayed at a distance from the nasus of the soldiers in Nasutitermitinae[47]. Other strategies are as peculiar as closing entrance holes with soldier heads (phragmosis) or strikes by modified symmetrical or asymmetrical snapping mandibles causing devastating wounds to invertebrates[34,41,54,55]. Defense is not restricted to soldiers, the defensive mechanism of workers in *Neocapritermes taracua* involves self-sacrifice through body rupture, allowing two separately stored secretions to come into contact together and to produce a sticky and toxic cocktail harmful to opponents[56,57].

In nature it is common to see individuals of some social insects warning their nestmates of potential danger. These direct observations repeatedly revealed the importance of alarm communication in many insects[23,37,58]. In termites, any disturbance triggers seemingly erratic movements leading to effective defensive responses[34–36,59,60], and enhanced protection of the colony due to the increase of the soldiers-to-worker ratio at the place of disturbance. This strategy makes the whole group unpalatable even to specialized vertebrate predators[61]. Our understanding of the proximate mechanisms of alarm communication is still limited, and because termites, ants, bees, and wasps are known to respond to the threat stimuli in a context-dependent fashion, the acquisition of empirical data and their interpretation remains challenging[62].

In spite of the crucial importance of alarm communication for termite colony survival, only fragmented reports have hitherto been published about this topic, most of which focused on either vibroacoustic or pheromonal communication of isolated species[27,36,59,60,63–71]. The evolutionary trajectories of alarm signals, and their significance within complex ecological constraints across extant termite lineages, have not previously been investigated, and there is no report on alarm communication in soil-feeding termites, which represents over half of termite diversity[70]. In this work, we carried out a detailed study on nine termite species that, combined with existing knowledge on alarm communication in cockroaches and termites, includes members of all major lineages and ecological strategies (Table S1). We then studied the evolution of alarm characters in a phylogenetic context, and in relation to a series of social and ecological features,

which are key factors influencing communication abilities in social animals[58].

While vibroacoustic alarm signals evolved prior to achieving eusociality, the occurrence of alarm pheromones is linked to a wood-feeding habit and to populous colonies. These results indicate how important these ecological features are for the means of communication within termite colonies. In spite of the strong evolutionary signal, the influence of environment is best evidenced in the desert termite *Hodotermes* having lost alarm signaling as it lives in sandy soils and forages outside. The sophisticated alarm communication of termites may at least partially explain the ecological success of these eusocial insects in the tropics.

## Terminology

We use the following definitions for the behaviors (all shown in Movie S1) observed in this study:

*Locomotion speed* is the average speed-of-motion of two individuals, either workers or soldiers, independently, per experiment, expressed in mm/s.

*Burst* is a sequence of oscillatory movements with stable spans between the beats at low or high frequency. It can be performed as a tremulation, drumming, or head-banging sequence.

*Tremulation* (body shaking) is the longitudinal oscillatory movements *sensu* Howse[71], during which the head or the abdomen rarely hits the substrate. Tremulation signals are used at either low (≤15 Hz) or high frequency (>15 Hz), which we refer to as "low tremulations" and "high tremulations", respectively.

*Drumming* is the vertical oscillatory movements *sensu* Howse[71] during which the abdomen hits the substrate. Drumming is present in both workers and soldiers and is always displayed at high frequency (>15 Hz).

*Head-banging* is performed exclusively by soldiers hitting the substrate with their heads at high frequency (>15 Hz).

## Results and discussion

**General behavioral responses (avoidance and aggregation).** Wilson and Regnier[38] classified the alarm responses in ants as either "panic" or "aggressive". As the specific alarm signaling in termites (comparable to panic in ants) is a subtle behavior[71,72] out of the scope of this work, general alarm is accompanied by a dramatic change in the group behavior. The general alarm typically involves many individuals disturbed at foraging sites, or present in a part of the nest that has been locally damaged[27,36,59,60,63–69]. The alerting termites search for quiet termites, touch them with their antennae, and perform tremulations to alert them[36]. The alarm responses usually result in a high soldier recruitment activity at disturbance locations, where soldiers displayed defensive postures, often combined with the release of defensive secretions produced by the labial glands during the opening/closing of mandibles[26,27]. Locomotion activity increased in many cases, especially in *Reticulitermes*, and disturbed workers usually displayed higher locomotion activity (evasion) than soldiers (defensive confrontation; Fig. 1 and Tables S1 and S2).

Workers attempted escaping from the source of disturbance by moving away rapidly, while soldiers often searched for the source of disturbance and aggregated around it, resulting in a slower-motion patrolling behavior usually combined with scanning of the space with wide-spread antennae and mandibles ready to be triggered (when present). Soldiers of the majority of the studied species with biting-type mandibles started opening mandibles after direct disturbance because of two reasons: (i) open mandibles are prepared to bite once the opponent reaches the antennae, (ii) repeated openings of mandibles stimulate the

release of defensive chemicals from cephalic glands as the mandibular muscles squeeze the liquids out of reservoirs. The defensive secretion is usually delivered to the opponent together with the bite[34,47,54,73]. The response to disturbance also included higher production of vibroacoustic alarm by disturbed individuals that warns nestmates (Table S3). Aside from such common behavioral responses, more specific actions were repeatedly observed. Workers and soldiers of *Hodotermopsis* frequently showed a type of drumming that substantially differs from that of all other termites—vigorous oscillatory movements against the lid of the Petri dish (not against the ground as in all other cases). While the increased locomotion was mainly observed in response to direct disturbance, there were a few exceptions, as in *Neotermes* and *Termes*, in which soldiers significantly decreased their locomotion speed, although such change was observed in *Neotermes* in long-term response only, implying that the patrolling behavior followed the active search for the source of disturbance. Finally, *Hodotermes* was a remarkable outlier in its general behavioral response to disturbances, as both castes stopped all movements after the disturbance for a short time (freezing behavior), but resumed their previous activity within 1–2 seconds. Thus, based on our experiments and repeated field observations, it is likely that the escaping behavior (higher locomotion speed) of workers and the aggregation of soldiers towards a disturbance is a basal trait to all extant termites that was secondarily lost only once, in *Hodotermes* (Fig. 1 and Table S1).

**Alarm pheromones**. Alarm pheromones in termites originate from soldiers' defensive glands only: the labial glands in *Mastotermes*[27] and the frontal gland in Neoisoptera (the derived group comprising Stylotermitidae, Rhinotermitidae, Serritermitidae, and Termitidae)[36,59,63–65,67–69]. Similar signals are widely used in some cockroaches, produced by the abdominal sternal or tergal glands (*Eurycotis*[74]; *Therea*[75]; *Blaberus*[76]). Benzoquinone, the alarm pheromone of *Mastotermes*, originates from the soldiers' labial glands, and triggers a typical alarm behavior including caste-dependent change in locomotion speed and increased vibroacoustic signaling. All other alarm pheromones originate from the frontal gland, a termite-specific organ with no equivalent in other groups[77]. The frontal gland of soldiers is a saccular gland that opens to the exterior through the fontanelle in all Neoisoptera species we studied but *Glossotermes*, which has a blind-ended sac in the thorax and abdomen, whose contents can eventually be released through self-sacrifice via body rupture[34,73].

Our study included three genera of Neoisoptera using alarm pheromones—*Prorhinotermes*, *Reticulitermes*, and *Constrictotermes* (Fig. 2 and Table S1), along with data from the literature. The active components of alarm pheromones are terpene hydrocarbons in all Neoisoptera[34,41,68]. Although *Glossotermes* increased the locomotion speed in response to a crushed soldier head, our chemical analyses did not reveal any alarm pheromone candidate (Fig. S1, Table S4), in line with the lack of a frontal gland reservoir in its head[73]. However, more species revealed responses to crushed heads devoid of defensive glands or conspicuous volatiles (see Fig. 1 and Table S1), possibly because colony members can perceive the smell of dead or wounded termites[78–81]. The two soil-feeding species we studied (*Labiotermes* and *Termes*) lack alarm pheromones, unlike wood-feeders that used alarm pheromones. In addition to field observations, it suggests that alarm pheromones are not used by the soil-feeding groups, representing altogether 60% of termite species diversity[70]. According to recent termite phylogenies[82,83], alarm pheromones evolved at least twice; in the most basal extant termite clade, Mastotermitidae, and then in Neoisoptera, in which the lack of

|  | Short-term response to disturbance | | | | | | | | Long-term response to disturbance | | | | | | | |
|  | Workers | | | | Soldiers | | | | Workers | | | | Soldiers | | | |
|  | Light | Air | CWH | CSH | Light | Air | CWH | CSH | Light | Air | CWH | CSH | Light | Air | CWH | CSH |
|---|---|---|---|---|---|---|---|---|---|---|---|---|---|---|---|---|
| *Cryptocercus* | ✗ | ↑ | ↑ | NA | NA | NA | NA | NA | ✗ | ↑ | ↑ | NA | NA | NA | NA | NA |
| *Mastotermes* | ✗ | ↑ | ✗ | ✗ | ✗ | ✗ | ✗ | ✗ | ↑ | ↑ | ↑ | ↑ | ✗ | ✗ | ↑ | ↑ |
| *Hodotermopsis* | ✗ | ↑ | ✗ | ✗ | ✗ | ✗ | ✗ | ✗ | ✗ | ✗ | ✗ | ✗ | ✗ | ✗ | ✗ | ✗ |
| *Hodotermes* | ✗ | ✗ | ✗ | ✗ | ✗ | ✗ | ✗ | ✗ | ✗ | ✗ | ✗ | ✗ | ✗ | ✗ | ✗ | ✗ |
| *Neotermes* | ↑ | ↑ | ✗ | ↑ | ↑ | ↑ | ✗ | ✗ | ✗ | ✗ | ✗ | ✗ | ✗ | ↓ | ✗ | ✗ |
| *Glossotermes* | ✗ | ↑ | ↑ | ↑ | ✗ | ↑ | ↑ | ↑ | ↑ | ✗ | ✗ | ✗ | ✗ | ✗ | ✗ | ✗ |
| *Reticulitermes* | ↑ | ↑ | ✗ | ↑ | ↑ | ↑ | ↑ | ↑ | ↑ | ↑ | ✗ | ↑ | ↑ | ↑ | ↑ | ↑ |
| *Labiotermes* | ✗ | ✗ | ✗ | ✗ | ✗ | ↑ | ✗ | ✗ | ✗ | ✗ | ✗ | ✗ | ✗ | ✗ | ✗ | ✗ |
| *Termes* | ✗ | ✗ | ✗ | ✗ | ↓ | ✗ | ✗ | ✗ | ✗ | ✗ | ✗ | ✗ | ✗ | ✗ | ✗ | ✗ |
| *Constrictotermes* | NA | NA | NA | NA | NA | NA | NA | NA | NA | NA | ✗ | ↑ | NA | NA | ✗ | ↑ |

**Fig. 1 Short-term (left) or long-term (right) responses in the locomotion speed in groups of nymphs of the wood roach *Cryptocercus*, or of workers and soldiers of different termite genera after exposure to different stimuli (light flash (light), air current (air), crushed worker head (CWH), and crushed soldier head (CSH)).** Locomotion speed was recorded separately for workers (*n* = 2) and soldiers (*n* = 2). The green fields indicate significantly different locomotion speed after stimulation, upward arrows mean the speed increased, downward arrows mean the speed decreased, the blue fields indicate no significant difference, NA means data not available.

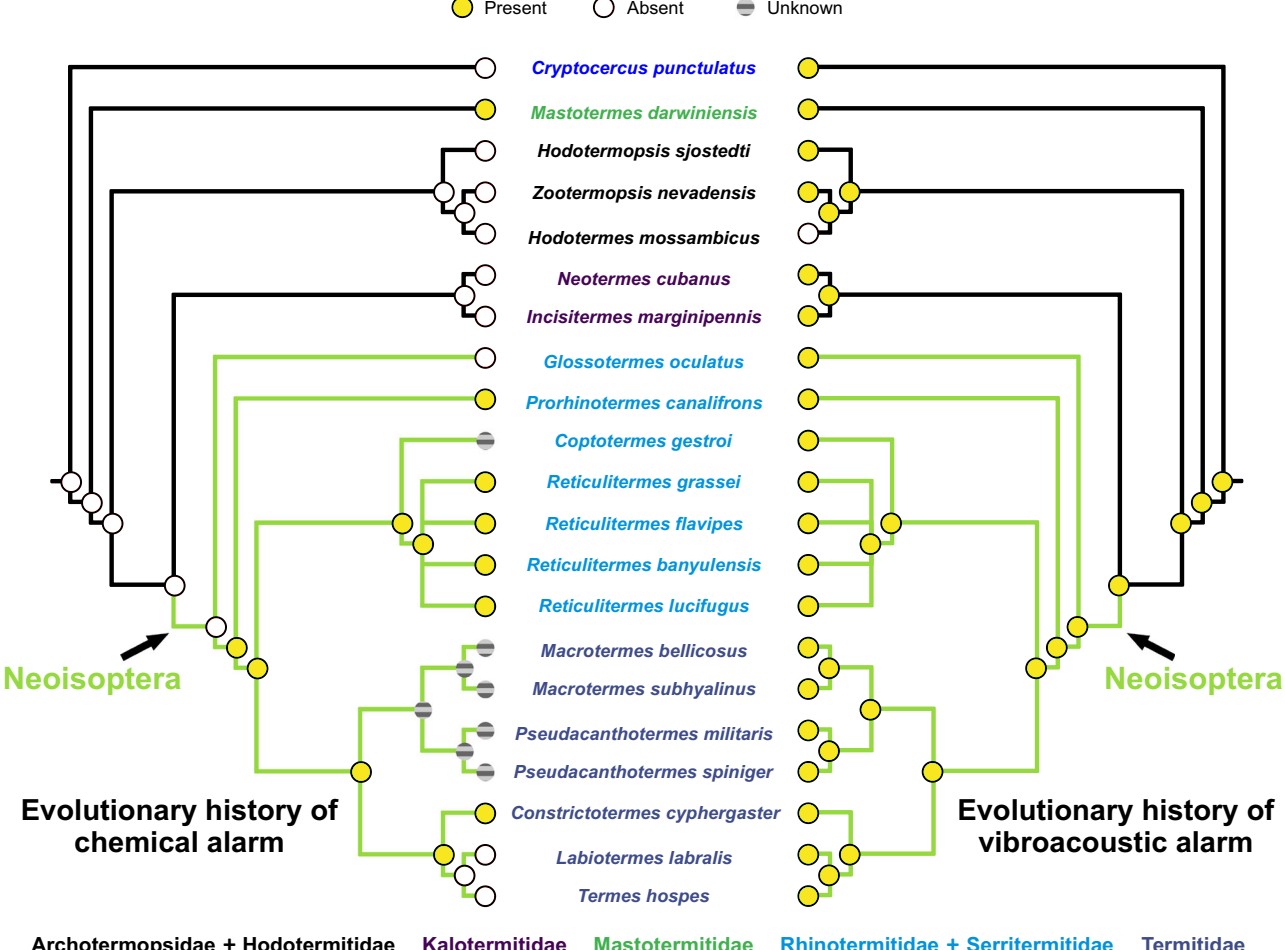

**Fig. 2 General comparison of the evolutionary history of vibroacoustic and chemical alarm distribution among termites and the wood roach *Cryptocercus*.** Dots are yellow when the alarm mode is present, white when absent, and gray when unknown. The family name colors represent ranks for MCA analysis.

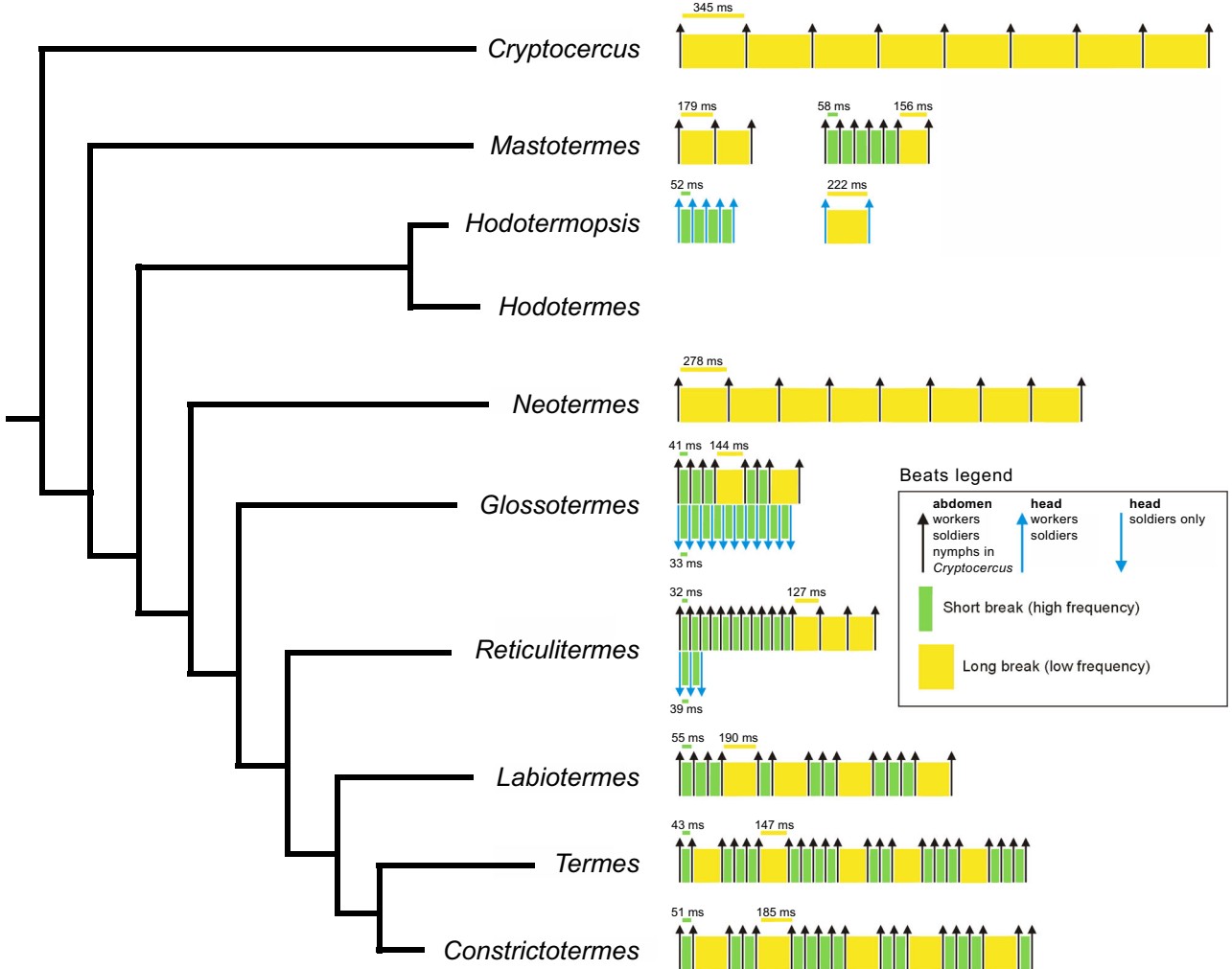

**Fig. 3 Scheme of the main features of the vibroacoustic alarm communication linked to phylogeny (*n* = 50 for each species and castes).** In contrary to Fig. 2, we included only species for which vibroacoustic communication is known in sufficient details.

observations precludes our distinguishing between a single origin followed by multiple losses or multiple origins (Fig. 2). Our investigation suggested that the species' life-type and related traits strongly correlated with the presence of alarm pheromones (Fig. S2), as species living either in small colonies (Archotermopsidae, Kalotermitidae[70,84]), or those living in open arid environments (Hodotermitidae[70,84]), lack this communication channel.

**Vibroacoustic signaling**. Termites generally responded to disturbance by violent shaking and drumming, sometimes accompanied by sounds audible to the observer (Movie S1). The beats were arranged into bursts of low frequency (under 15 Hz) in the case of low tremulation, or high frequency (above 15 Hz) high tremulation, drumming or head-banging. The vibroacoustic signature was specific to a given genus (Figs. 3, 4, and S3), a feature not previously recognized.

While the species descending from early diverging lineages (including *Cryptocercus*[85]) revealed a rather monotonous pattern of beats arranged into singular bursts, the patterns became more diverse in Neoisoptera, such as in *Glossotermes*, and especially in Termitidae, which use a combination of several bursts into a single vibroacoustic event (Figs. 3 and S3). The tremulations were primarily used as short-range communication to alert naïve

nestmates, and carried relatively low energy compared to drumming or head-banging. The tremulations, when processed and amplified, are audible as muffled noise, while drumming and head-banging sounds like a series of sharp hits (Movie S1). The occurrences of respective signal components are summarized in Table S1. Workers and soldiers within a species mostly share the same repertoire, although additional signals, such as head-banging, occur in soldiers only. The general trends show that the larger species vibrate at lower frequencies, and *Cryptocercus* and *Neotermes*, both high above the average termite size, lack the high-frequency signals completely. *Hodotermopsis* is unique among termites for hitting its head against the ceiling for drumming, not the floor, in both workers and soldiers. Head-banging was abundantly recorded in *Glossotermes* and *Reticulitermes* (Figs. 3, 4, and S3), but was only rarely observed in *Labiotermes*. All Termitidae displayed relatively conserved and complex patterns of vibroacoustic sequences combining abdomen drumming, and high and low tremulations into long sequences (Movie S1, Figs. 3, and S3). Notably, vibroacoustic signals were extremely well conserved within species, with little variation among subsequent beats, showing that alarm signals have been quite stable since *Cryptocercus* and termites diverged (Figs. 5 and S4), an event dating back to at least the Late Jurassic[82,83]. We may therefore assume that the pre-social ancestor of *Cryptocercus* and termites used alarm signals of comparable precision.

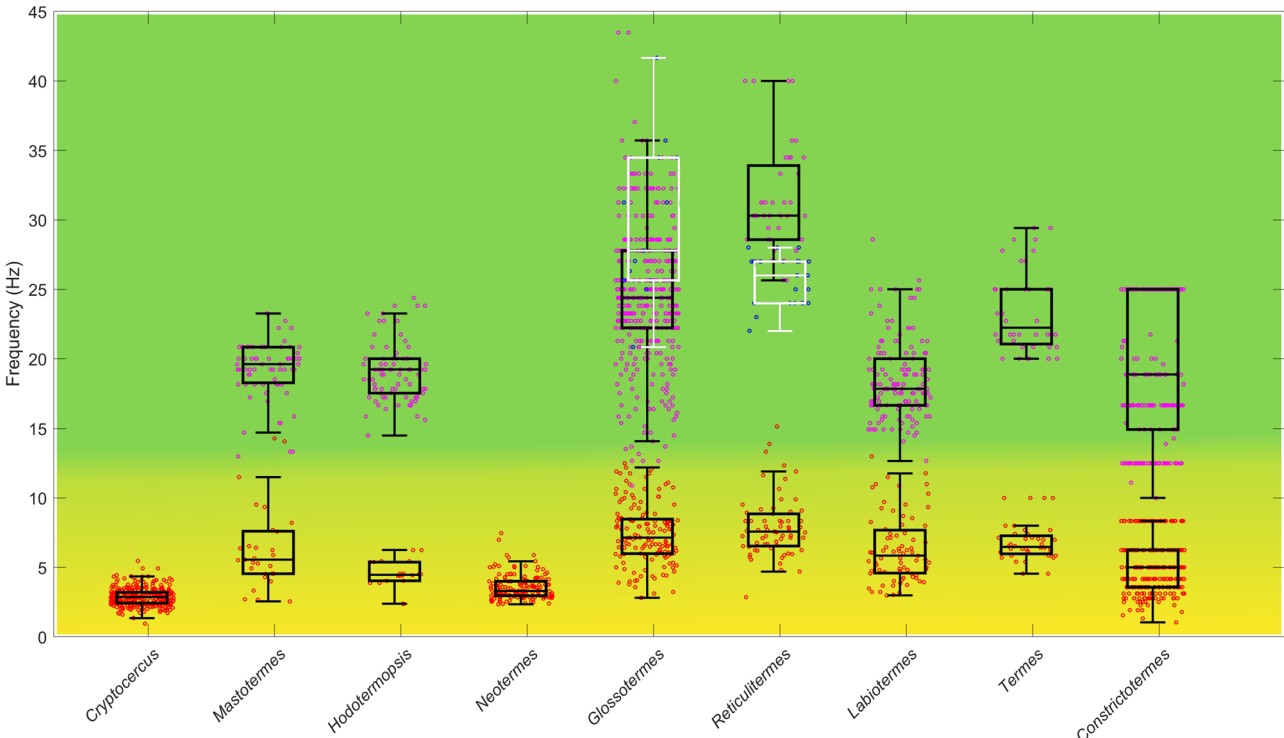

**Fig. 4 Frequency (in Hz) of low tremulations (yellow background) and high (green background) tremulations (*n* = 50 for each species and castes) across studied species.** The nymphs of the wood roach *Cryptocercus* and termite workers in black, and drumming by termite soldiers (in white). On each box, the central mark is the median, the edges of the box are the 25th and 75th percentiles, the whiskers extend to the most extreme datapoints the algorithm considers to be not outliers, and the outliers are plotted individually. *Cryptocercus* and *Neotermes* produced low frequency signals only. *Hodotermes* is not included as it does not communicate via body vibrations.

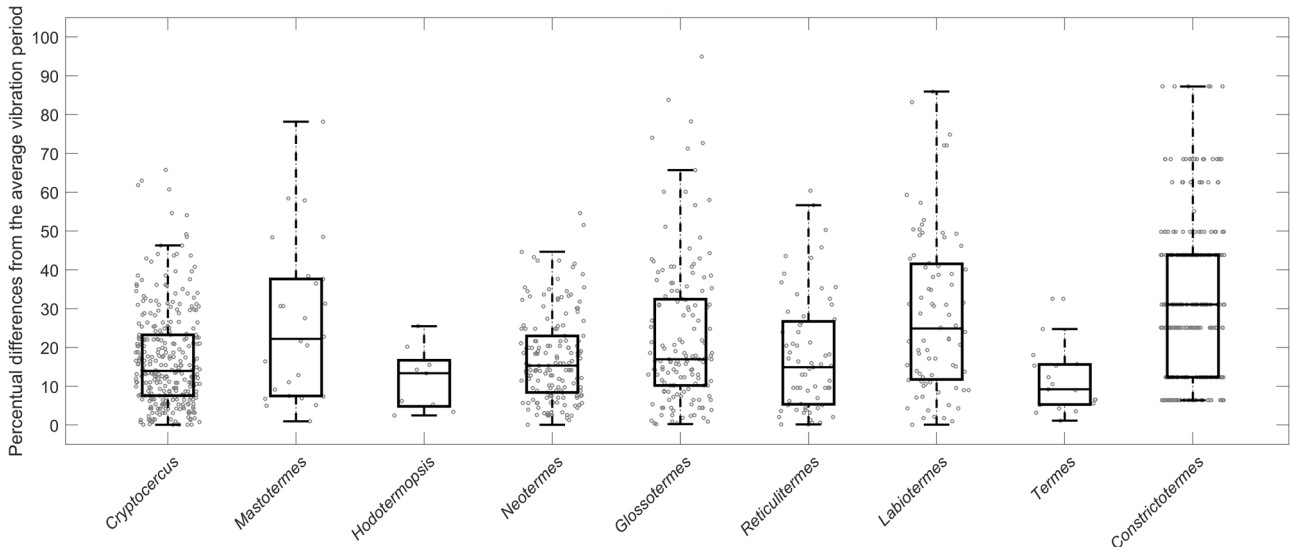

**Fig. 5 Stability of breaks between two subsequent beats in low tremulations across the species (*n* = 50 for each species).** The values are given as percentage difference (positive or negative) from the mean duration. On each box, the central mark is the median, the edges of the box are the 25th and 75th percentiles, the whiskers extend to the most extreme datapoints the algorithm considers to be not outliers, and the outliers are plotted individually.

**Evolutionary trajectories of alarm signals in termites**. Our ancestral-states reconstructions indicate a single origin of vibroacoustic alarm communication in the common ancestor of all termites and their sister group, the wood roach *Cryptocercus* (Fig. 2). Its loss in *Hodotermes* (and *Anacanthotermes*, D.S.D. and J.Š., field observation) is probably due to environmental conditions: nesting in soft sandy ground and randomly foraging in the open air to collect dry grass presumably prevents effective

transmission of vibrations or odors. To our best knowledge, the loss of vibroacoustic alarms occurred exclusively in Hodotermitidae. Whether vibratory communication disappeared completely in this group, or whether it was partly retained in another social context such as nest defense, as it is common in other termites[86–89], remains to be determined.

Even though vibroacoustic communication is shared by all colony members, the actual involvement of the different castes in

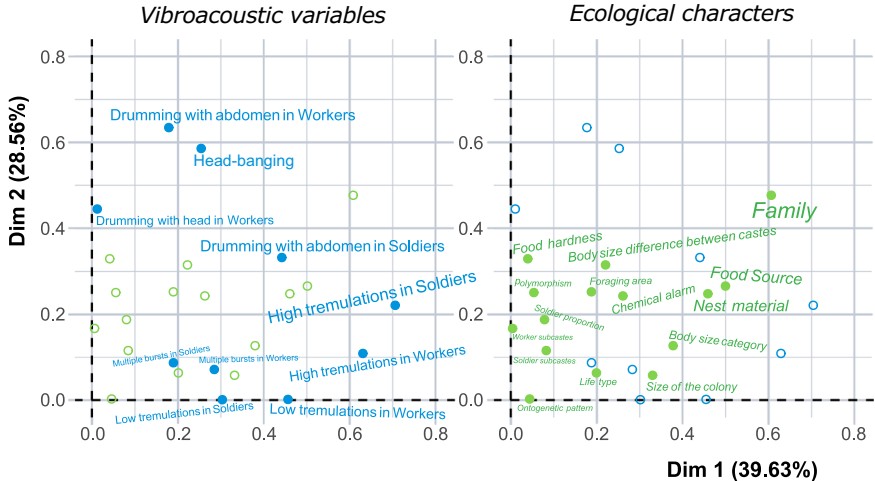

**Fig. 6 Square correlation ratio between characters and axis in MCA.** Both graphs are in fact the same, and the left one shows the most influential characteristics of vibro-acoustic features (in blue) to the MCA analysis and the right one shows the most important ecological characters (in green) that might have an effect on vibroacoustic traits. The larger the size of the letters, the more important the character.

alarm signaling has rarely been studied. Our data show that both castes mostly share identical part on the communication, except in *Glossotermes*, *Reticulitermes*, and *Labiotermes*, in which soldiers perform head-banging, a soldier-specific behavior. Apart of these differences, our data do not support increased involvement of soldiers in alarm signaling, in contradiction to observations made by Stuart[90]. We could observe increased diversity of alarm signals in derived termite taxa, evidenced mostly in vibroacoustic alarm sequences, since alarm pheromone data are made only in the presence/absence of this channel.

Multiple Correspondence Analysis (MCA) analysis suggested that several ecological characteristics have a strong influence upon vibroacoustic communication. The most prominent ecological characters are the hardness of the food and the nest material, which, along with phylogenetic position, are the most important features influencing vibroacoustic signaling (Figs. 6 and S5), probably because relatively hard substrates facilitate the transmission of such signals. In addition, we found a clear relationship between the frequency of oscillatory movements used in vibroacoustic communication and termite body size, since larger species always communicate at lower frequencies than smaller ones.

Pairwise correlation analyses showed that vibroacoustic characters are strongly correlated between the soldier and worker castes. Moreover, it seems that the presence of tremulation correlates with drumming in both castes across all species, and that the presence of tremulation in the soldier caste is correlated with the size ratio between castes (Fig. S2). Termite taxa with proportionally large soldiers compared to workers, such as *Hodotermopsis* or *Neotermes*, primarily rely on soldier behavior for the spread of alarm, while termite taxa in which the soldier and worker castes have similar sizes rely equally on soldiers and workers to communicate. There is certainly a phylogenetic component to this as soldiers in more basal lineages tend to be larger, but it possibly also reflects the fact that heavier individuals spread vibroacoustic alarm more efficiently, rendering the spread of alarm communication by smaller individuals obsolete. Whether the specialization for alarm transmission of larger individuals holds or not among subcastes of species with polymorphic soldiers, such as some Macrotermitinae (*Macrotermes*, *Acanthotermes*, *Ancistrotermes*) or some Nasutitermitinae (*Diversitermes*, *Trinervitermes*), remains to be investigated.

An interesting correlation exists between the presence of chemical alarm and the colony life-type (Fig. S2). One-piece

nesters living in hard wood (common among basal termites) do not utilize chemical alarms, probably because vibroacoustic alarm communication is sufficient in small colonies sheltered in sound wood[84,91]. These species usually defend against intruders at a few "bottleneck" entrance points in the gallery network[92]. Termite colonies contained within a small gallery system of a single piece of wood also use vibrations for purposes other than alarm communication. For example, *Cryptotermes* spp. may evaluate a looming resource shortage and perceive approaching competitors through substrate vibrations, which may result in the initiation of mass production of alates for a final dispersal flight[30,31]. Our data suggest that chemical alarms have only emerged in termite species that are able to use food resources outside their nest location, although not all central site-nesting termites appear to possess these. Species colonizing new food sources through underground foraging galleries are more likely to encounter enemies inside their galleries[93], potentially increasing the selection pressure for the acquisition of an alarm pheromone. The use of volatile alarm pheromones may efficiently alert naïve individuals of a potential threat when they approach a disturbed area, while soft substrates may not favor vibroacoustic communication. We can speculate that alarm pheromones may persist in the air even after the death of the foraging termites, triggering local avoidance of the newcomers.

The identification of alarm pheromones is intricate, and therefore they have been identified for only a few species[27,36,59,65,68,94]. Here, we have not studied the chemical nature of alarm pheromones, but merely their presence/absence based on the soldier crushed heads, behavioral responses to them, and the composition of their volatiles. A maximum parsimony model supports independent evolutionary origins of chemical alarms in Mastotermitidae and Neoisoptera, followed by repeated losses in the latter lineage. Many termite and cockroach species produce defensive compounds, often irritating, that, presumably, can be co-opted as alarm pheromones, explaining the diversity of alarm pheromones and their glandular origins found among Blattodea under highly variable selective pressures[27,34,74–76].

**Conclusion and future directions**

Alarm behaviors are ubiquitous in termites (excepting Hodotermitidae), confirming that caste-dependent responses to disturbances (workers primarily hide away while soldiers confront the threat) is a plesiomorphic characteristic of termites that later

diversified with the rise of extant termite lineages. The use of vibroacoustic alarm signals evolved prior to the evolution of eusociality in termites, as their homologs are present in the wood roach *Cryptocercus*, indicating a shared origin in their most recent common ancestor. Subsequently, as termite lineages proliferated, the nature of vibratory signals became progressively more variable in Neoisoptera, with clear patterns of low and high tremulations, drumming, and head-banging. Alarm pheromones appeared in soldiers at least twice, from compounds secreted by the labial glands in *Mastotermes*, or from compounds derived from the frontal gland in Neoisoptera. However, while the soldier frontal gland was a major evolutionary innovation that likely contributed to the success of Neoisoptera[48,70], the allomonal secretions may have gained the secondary function of an alarm signal in some clades only. The absence of alarm pheromones in *Glossotermes* therefore raises the possibility that the common ancestor of all modern Neoisoptera did not use an alarm pheromone, which, instead, first evolved later on, in more derived Rhinotermitidae, and was secondarily lost in some taxa (*Labiotermes*, *Termes*, etc.). The investigation of phylogenetically basal Neoisoptera (Stylotermitidae, Rhinotermitinae) is needed to confirm, or reject, this scenario. Although *Mastotermes* is the only non-Neoisoptera known to use an alarm pheromone, it is still possible that some other understudied taxa have acquired this chemical signal as well. For example, *Paraneotermes simplicicornis* is the only member of Kalotermitidae known to have the ability to nest underground and to forage for many wood items[95,96], a trait we found strongly associated with the use of a chemical alarm. The investigation of such outliers may provide additional insights into the evolution of alarm behaviors, and into the ecological pressures driving them. Another interesting factor possibly influencing alarm communication is the presence of a soldier caste, which was lost at least three times independently in (i) Apicotermitinae, (ii) *Orientotermes* and *Protohamitermes*, and (iii) *Invasitermes* (all Termitidae[97–99]). Workers in these groups are fully responsible for colony defense, and they thus reveal high levels of agonism and sometimes also developed unique defense strategies, such as body rupturing[34,47,100]. Furthermore, the remarkable absence of alarm signals and responses to danger stimuli in Hodotermitidae underline the effect of ecological factors on the communication skills in a given species. The investigation of other termite species with various feeding, foraging, and nesting habits could therefore reveal novel defense mechanisms, including alarm communications that may have been selected for under different ecological pressures. Regardless, communication among individuals responding to distress evolved well prior to the eusocial system so characteristic of termite life and from vibroacoustic systems found widely among arthropods. Subsequently, chemical refinements to this communication system evolved multiple times and assuredly contributed to the considerable Cenozoic radiation of Neoisoptera, principally Termitidae, and their ecological dominance of tropical ecosystems[101]. Future research should, among other avenues, focus on fine comparisons of the alarm communication between termites and ants, or more generally speaking amongst all eusocial groups. These insects share common patterns of social organization, and ants have already been studied in respect to alarm communication in considerable detail[23,102–104].

## Material and methods

**Biological material**. Representative species from most major termite taxa, in addition to the wood roach *Cryptocercus punctulatus*, a member of the extant sister group to termites[82,105,106], were used by using one colony for each in this study (Supplementary Data 1). Information from previous publications were reanalyzed and standardized to increase dataset coverage across species, and a newly acquired dataset of termite species was obtained from laboratory colonies and/or field colonies, following the protocols described below. All material was transported to Prague (Czech Republic) following legal procedures with the full array of permits from the country of origin. The combined dataset allowed for a comparative analysis of the evolution of alarm communication components in each termite taxon, which we linked with ecological or developmental traits. A detailed description of material origin, ecological and developmental traits, and methodological approaches is provided in Supplementary Data 1.

**Behavioral experiments**. The experimental groups consisted of workers and soldiers maintained in their species-specific caste ratio (according to[107]; for details on caste ratio, see Supplementary Data 1) in a 85 mm Petri dish[108]. Only the species whose caste ratio is indicated in Supplementary Data 1 were studied. Tested stimuli consisted of (A) light—flash of three seconds (800 lux, 5500 – 6000 K color temperature), (B) air current—3 s of human breath through a fine straw to mimic a breach into a nest, (C) one crushed worker or soldier head spread on a piece of filter paper (CWH or CSH, respectively). (A) and (B) are hereafter called "direct disturbances". All experimental groups were introduced in a Petri dish, and left for two hours undisturbed before being exposed to one of the disturbance stimuli. Stimuli and controls were replicated six times on independent groups by a single person, and recorded in full HD with Canon EOS 6D combined with EF 100 mm f/2.8 L Macro IS USM. Each video was recorded for a total of seven minutes, including one minute before the introduction of the stimulus, and six minutes after the stimulus was introduced. We then analyzed the short-term (1 min) and long-term (6 min) response of termites. The locomotion speed of two workers and two soldiers selected randomly (we used both soldiers in an experiment in species with a low soldier-to-worker ratio) were obtained from each replicate by using Mouse-Tracer software (ref. [36]) and allowed us to learn about the presence or absence of an alarm pheromone in each species based on responses to CSH.

**Chemical analysis**. Substances that could be alarm pheromones were investigated in all focal species by chromatographic analysis. Cold-anesthetized termites were dissected using a stereomicroscope. Termite heads with the frontal gland from 2 to 20 individuals were placed into a 2-mL clear glass vial, crushed using a glass rod and the vial was closed with a PTFE/silicone septum cap. The headspace extraction of volatiles was carried out using an SPME fiber holder for manual sampling equipped with a fused silica fiber coated with 30 μm polydimethylsiloxane (Supelco, Bellefonte, USA). The holder needle was passed through the vial septum and the fiber was exposed for 10 min at room temperature. The analytes were desorbed at 220 °C in a split/splitless injector of a 5975B quadrupole mass spectrometer coupled to a 6890 N gas chromatograph (Agilent, Santa Clara, CA). The separation was achieved on a DB-5ms capillary column (30 m × 0.25 mm, a film thickness of 0.25 μm, Agilent) at a constant flow mode (1 ml/min) with helium as a carrier gas. The temperature program was: 40 °C (1 min), then 5 °C/min to 200 °C, then 15 °C/min to 320 °C (3 min). The temperatures of the transfer line, ion source, and quadrupole were 280 °C, 230 °C, and 150 °C, respectively. The compounds were ionized by 70 eV electrons.

Moreover, the tissue compounds were extracted with a small amount of hexane, i.e. the liquid extraction was achieved by adding 50–100 μl of hexane to freshly crushed termite heads. The extracts were analyzed on the same GC-MS instrument and the

same column as above with slightly modified parameters. The injector held at 250 °C and operated with a split ratio of 1:20 injected 1 μl of the extract. The temperature program was as follow: 50 °C (1 min), then 15 °C/min to 200 °C, then 6 °C/min to 320 °C (3 min); total run time was 34 min. Data were recorded with a 4-min solvent delay. Detailed data on the identity of alarm pheromones originated from previous works of our team[27,68,69].

**Vibroacoustic experiments**. The experiments were carried out on all the species indicated in Fig. 4 in an anechoic room, which provides low background noise. Prior to experiments, the experimental design of each 85 mm Petri dish (distance between floor and ceiling, coursing the floor, humidity, etc.) was optimized for each species. Videos were recorded with a Panasonic HDC-TM700 camera placed over the testing arenas, which allowed us to trace the origin of particular vibro-acoustic behaviors for each species and caste. The vibroacoustic recording system comprised high-sensitivity accelerometers (Brüel & Kjær type 4507 B 005) fixed on the bottom or on the lid of the Petri dish according to the species' behavior. The accelerometric data were analyzed as overall alarm energy derived from the sum of tremulation and drumming after stimulation, normalized to the status before stimulation. The characteristics, nature, and frequency of beats composing bursts were analyzed until 50 observations were completed for each behavior. The entire records were analyzed in cases where less than 50 behavioral observations occurred. For the detailed procedure of data acquisition, see Fig. S6.

**Reconstruction of ancestral states**. We reconstructed the ancestral states of 33 characters (see Table S1) for 21 species using a phylogenetic tree of termites inferred from full mitochondrial genomes[82,109]. Alarm characters were selected from our observations and existing literature, ecological characters were selected according to our estimation of possible importance. The nature of the diet (column X31) was indicated according to existing literature[110] and field observations. We used "Maximum parsimony" and "Maximum likelihood" methods, build-in Mesquite software (v3.6[111]), to estimate characters' ancestral states. Because some analyses could not run with empty data, the characters with no values were removed prior the analyses.

**Statistics**. The behavioral experiments were evaluated using the Kruskal-Wallis test and post-hoc two-by-two permutation tests for independent samples (P values). The Bonferroni-Holm correction[112] was applied for multiple comparisons among different conditions (H values). Accelerometric data were compared using t-tests for paired samples. All statistics were performed using StatXact (Cytel Studio, version 9.0.0, 2010) and SigmaPlot software (Systat Software Inc., version 11.0.0.77, 2007).

**Character correlation analyses**. To assess characters' evolutionary correlations, we carried out a three-step approach. First, we performed statistical correlation using Chi-square test with p-value simulated by 500,000 iterations (chisq.test function in R 3.5.2.). Significant correlations were arbitrarily set to $p < 0.01$, to avoid random effects as much as possible. Secondly, we performed Pairwise Comparisons contrasting in state of two characters[113] implemented in the software package Mesquite (v3.6[111]). In all cases, we always selected the Pairwise model with the highest "best tail" p-value for further scoring, and only scores above 0.8 were analyzed further. We scored every character pair in our dataset using the following equation:

$$PC = PW * R \qquad (1)$$

in which

$$PW = \frac{(Pos - Neg)^2}{nPW} \qquad (2)$$

and

$$R = \frac{\frac{1}{PW} + nR^+}{1 + nR + \frac{nR}{2}} \qquad (3)$$

$PW$ is the coefficient of taxa pairs in Pairwise comparison, based on difference between positive ($Pos$) and negative ($Neg$) pair sets in the pairwise model. $nPW$ is the number of pairs in model. $R$ is the coefficient of remaining taxa/nodes not included in any pair. $R$ is expressed for all nodes, excluded from pairs connections defining clades with the same character states to avoid "pseudoreplication of lineage-specific factors"[114]. A clade is then defined as a separate node including terminals, if the sister clade differs in the given character state. $nR^+$ is the number of nodes supporting $PW$, $nR^-$ is the number of nodes contradicting $PW$. $nR$ is the total number of nodes not included in clades as defined above. Taxa containing unknown states in a particular character pair were omitted from the analysis. Lastly, we selected pairs of characters fitting both criteria and created the mirror trees (Mesquite software, v3.6[111]) to display the correlations.

**MCA analysis of ecological traits effect on vibroacoustic communication**. To determine the effects of environment and species ecology on means of vibratory communications, we performed Multiple Correspondence Analysis (MCA, libraries "FactoMineR" and "MissMDA" in R software, v3.5.2, https://www.r-project.org/). 14 characters (X21–X34, see Supplementary Data 1) were used as supplementary variables to 10 independent vibroacoustic alarm characters (X5–X14, see Supplementary Data 1).

**Reporting summary**. Further information on research design is available in the Nature Portfolio Reporting Summary linked to this article.

**Data availability**

All data generated or analyzed during this study are included in this published article (and its supplementary information files). The raw data for Figs. 4, 5, and S4a, b is included in Supplementary Data 2–5.

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

## Acknowledgements

We acknowledge our late friend, Philippe Cerdan, and also Régis Vigouroux and all Hydreco staff for their hospitality, and to František Jůna for recording behavior of *Cryptocercus*. We are grateful to Christine Nalepa (North Carolina State University, USA), who kindly provided us several family groups of *Cryptocercus punctulatus* used in this work, to Reinhard Leuthold who provided *Hodotermes mossambicus*, and to Alain Robert who provided *Termes hospes* and *Hodotermopsis sjostedti*. This study was supported by project CIGA No.2015430 (Czech University of Life Sciences Prague), by Faculty of Tropical AgriSciences (project IGA 20223112), and USDA National Institute of Food and Agriculture Hatch projects number FLA-FLT 005660. We also thank the anonymous reviewers and the editor for their helpful and constructive comments. Permits: Part of our material originated from colonies maintained in the lab since decades, when no specific permissions were required to import living material to Europe. *Hodotermopsis sjostedti* was collected by Alain Robert in Tam Dao forest (Vietnam) in 1990. *Hodotermes mossambicus* was collected by Reinhard Leuthold in Kenya in 1990. *Neotermes cubanus* was collected by Ivan Hrdý in Topes, de Collantes, Siera de Escambri (Cuba) in 1988. *Glossotermes oculatus* and *Labiotermes labralis* have been collected under permit approved by the French Ministry for the Ecological and Solidarity Transition (UID: ABSCH-CNA-FR-240495-2) (TREL1902817S/118). *Termes hospes* was collected by Alain Robert near Pointe-Noire in the Republic of the Congo. *Cryptocercus punctulatus* was generously provided by Christine Nalepa (North Carolina State University, USA). No new material of *Constrictotermes cyphergaster* was used in this study, but instead, we have re-analyzed available the data published in Cristaldo et al. 2015[68], where the details of legal procedures are provided. The same is true for *Mastotermes darwiniensis* and *Reticulitermes flavipes*, for which the details are available in Delattre et al. 2015[27] and Delattre et al. 2019[69], respectively.

## Author contributions

D.S.D., formulation of research strategy, supervisor of behavioral part, writing the M.S.; V.J., vibroacoustic recording and signal processing, data analysis, writing the MS; P.S., statistical and ecological modeling; O.D., behavioral tests with termites, compiling the literature sources; T.C., writing the MS, data analysis; O.B., behavioral tests with *Cryptocercus*, evaluation of behavior; J.C., chemical analyses; D.S., behavioral results evaluation; J.S., preparation of experiments, behavioral results evaluation; M.B. and O.J., evaluation of vibroacoustic data; M.S.E. and T.B., results presentation, text edits; J.Š., formulation of research strategy, funding, coordination of all activities, writing the MS.

## Competing interests

The authors declare no competing interests.
