## [Peer Review File · Communications Biology]

Reviewers' comments:

Reviewer #1 (Remarks to the Author):

This is an interesting manuscript on an aspect of termite defence and communication that is currently understudied, and I thoroughly enjoyed reading it. The authors studied the vibroacoustic and chemical communication of a variety of termite species and their sister taxon *Cryptocercus* as well as investigated the evolution of alarm signals within a phylogenetic context and how social and ecological variables have impacted the evolution of communication within termites. Whilst the data presented are very interesting and likely worthy of publication in *Nature Communications*, there are several issues with the paper, as discussed below, that need to be addressed.

Abstract:

Generally well written however could benefit from adding information on the knowledge gaps and why you are conducting this research before going into what you studied (Line 62). Additionally, concluding the paragraph on the implications of this research and framing it in the context of animal weaponry and defence on a broader scale will improve this section.

Lines 65-66: overly wordy sentence with a lot of jargon, consider rephrasing for clarity

Line 66: insert "have" between alarm and evolved

Significance statement

It is still not clear from this section why you have done this study and why this particular study is significant. Be explicit in why you think this research is important in a broad context.

Introduction:

The introduction is overly verbose, and full of jargon that has not been defined. Terminology is inconsistent throughout (e.g. what is 'alarm signalling'? And is it the same as 'alarm behaviours' and 'alarm communication?'). Consider sticking to one well-defined term rather than changing throughout. Additionally, references are sparse throughout the introduction. Given that alarm communication and vibratory signals are ubiquitous in the animal kingdom, it would be beneficial to begin the introduction by framing it in this context and giving examples, rather than diving straight into alarm communication in social animals.

Line 86: first sentence is poorly worded. Consider removing completely or rephrasing

Line 87: the word "mechanism" doesn't seem appropriate in this context-implies proximate causation- reword

Line 86-94: no references for the claims made. Remove or add appropriate citations.

Line 88-91: sentence is overly wordy, with a lot of jargon that makes it hard to read.

Line 94: define what "alarm signalling" is

Line 96 and 99-100: are "warning signal" and "alarm behaviours" the same as "alarm signalling"? Consider defining what each means or use a consistent term throughout.

Line 103-104: this sentence seems to be incomplete and out of place at the end of this paragraph. Might be better suited at the start of next paragraph

Line 104: It would be great to see alarm signals discussed in a broader context at the end of the first paragraph. Many animals utilise alarm communication, not just social insects. Consider talking about this and include specific examples before narrowing down to alarm communication in termites.

Line 105-107: need references

Line 110: odd word choice "unforgettable"; unforgettable by who?

Line 110: "many personal observations": be specific as whose observations you were referring to

Line 107-114: would have been great to see the inclusion of some of the other amazing termite weapon types (e.g. nasutes, symmetrical and asymmetrical snapping mandibles etc.)

Line 117: define "vibroacoustic communication"

Line 119: define "mechanical signalling"

Line 119-122: very long sentence. Consider splitting into 2 separate sentences

Line 123: remove "the" at start of sentence

Line 128: remove "the" before "vibroacoustic"

Line 130: remove "the" at start of sentence

Line 136: "direct observations"- vague. More detail needed

Line 137-140: long-winded sentence- consider splitting into 2 or rephrasing

Line 157: could include definitions for terminology used in the introduction here (e.g. vibroacoustic signalling, alarm signalling etc)

Results and Discussion

Figures could be improved by removing grey background (takes away from the main visual features of the figures). Colour scheme in Fig 4 should be changed as the black and blue colour of the boxplots can be hard to distinguish. All figures also have 2 captions, so double check which one should be included in the final manuscript. Fig 6 is missing an axis label on the x axis (vibroacoustic variables plot).

As my experience with phylogenetics is limited, I am unable to comment on this aspect of the paper, however, it would be beneficial if you could include how you created the phylogenies used in Figure 2 and 3.

Line 178-179- remove "the" in front of general alarm

Line 185-187: not convinced by this statement "locomotion activity increased in many cases"- use explicit numbers here because from figure 1, it looks like changes in locomotion activity were not significant most of the time.

Line 189-196: you define what the green fields and arrows mean, however you also need to define what the blue boxes with a x are to make things clear.

Line 198-206: very interesting observation of termite defensive behaviours!

Figure 2: How was the tree in this figure produced?

Line 246: include "taxa" or "genera" after Neoisoptera

Line 373-375: need references

Conclusions/future directions

Whilst the authors have summed up the results of the study well, it would be nice to see more about the implications of this research as well as expanding on the future directions of this field as they are only briefly touched upon.

Methods

The methods are somewhat lacking in detail and clarity, with important aspects missing from the section (i.e. how phylogenetic trees used in figures were constructed; sample sizes for behavioural experiments; how treatments were randomised etc.).

Line 446: include permit details

Line 454: sample sizes needed for experimental groups

Line 456-459: were treatments carried out by the same person each time? Variability in researcher technique when carrying out these treatments may influence results

Reviewer #2 (Remarks to the Author):

The present paper aimed at the evolution of alarm (vibroacoustic and chemical) communication in termites, comparing 20 termite species and a wood roach, *Cryptocercus*. The multidisciplinary approach applied in this study was important to explain the multiple characters that may be evolved in the evolution of these termite defensive strategies. The results indicate that vibroacoustic signals are a synapomorphy of termites and *Cryptocercus*, i.e., there is a shared origin in their most recent common ancestor. On the other hand, the authors suggest that chemical signals evolved at least twice in termite species.

Given the complexity of termite defensive behaviors, this research is quite interesting and may be useful for future phylogenetic and ecological investigations. The discussions and conclusion are robust and provide very important information for understanding the evolution of alarm communication in termites. However, I have few comments to be addressed by the authors prior to publication.

Among the 20 termite species studied, none includes the soldierless termite group. I think that will be interesting to include at least one Apicotermitinae species in discussion, since these termites have peculiar defensive strategies.

In Material and Methods, it is important to be clear which species were studied in behavioral and vibroacoustic experiments.

Line 516 – 500,000 interactions (not “interactions”).

I hope that my comments will be useful for the improvement of the manuscript.

Reviewer #1 (Remarks to the Author):

Abstract:

Generally well written however could benefit from adding information on the knowledge gaps and why you are conducting this research before going into what you studied (Line 62). Additionally, concluding the paragraph on the implications of this research and framing it in the context of animal weaponry and defence on a broader scale will improve this section.

=> Unfortunately the abstract must be 150 words or fewer according to the guidelines of the journal and we already reached the limit. Therefore, we cannot add additional information.

Lines 65-66: overly wordy sentence with a lot of jargon, consider rephrasing for clarity

=> We splitted the sentence into two sentences and we rephrased it slightly. We hope it helped.

Line 66: insert "have" between alarm and evolved

=> Done.

Significance statement

It is still not clear from this section why you have done this study and why this particular study is significant. Be explicit in why you think this research is important in a broad context.

=> We improved the text for this two aspects.

Introduction:

The introduction is overly verbose, and full of jargon that has not been defined. Terminology is inconsistent throughout (e.g. what is 'alarm signalling'? And is it the same as 'alarm behaviours' and 'alarm communication?').

=> We have significantly changed the Introduction in order to make it easier to follow. It is now explained at the line 101 that it is a defensive strategy and more details about alarm signals are provided in the next sentences. Alarm behavior has been defined in the introduction. Alarm communication may be considered as identical to alarm signaling and this information has been added in the text (the latter focus more on the signs themselves).

Consider sticking to one well-defined term rather than changing throughout.

=> Done.

Additionally, references are sparse throughout the introduction. Given that alarm communication and vibratory signals are ubiquitous in the animal kingdom, it would be beneficial to begin the introduction by framing it in this context and giving examples, rather than diving straight into alarm communication in social animals.

=> We improved the beginning of the introduction by discussing about alarm communication in animals, and in particular about alarm calls, common in mammals and birds. Many references have been added as well.

Line 86: first sentence is poorly worded. Consider removing completely or rephrasing

=> This sentence has been rephrased.

Line 87: the word “mechanism” doesn’t seem appropriate in this context-implies proximate causation- reword

=> The word “mechanism” has been replaced by “strategies”.

Line 86-94: no references for the claims made. Remove or add appropriate citations.

=> Appropriate references have been added.

Line 88-91: sentence is overly wordy, with a lot of jargon that makes it hard to read.

=> This sentence has been changed, we hoped it is easier to read now.

Line 94: define what ‘alarm signalling’ is

=> See answer above.

Line 96 and 99-100: are “warning signal” and “alarm behaviours” the same as “alarm signalling”? Consider defining what each means or use a consistent term throughout.

=> The term “warning signal” has been removed from the text. The term “alarm behaviour” is now defined.

Line 103-104: this sentence seems to be incomplete and out of place at the end of this paragraph. Might be better suited at the start of next paragraph

=> The sentence has been completed and it now starts the next paragraph.

Line 104: It would be great to see alarm signals discussed in a broader context at the end of the first paragraph. Many animals utilise alarm communication, not just social insects. Consider talking about this and include specific examples before narrowing down to alarm communication in termites.

=> We have discussed alarm signals in a broader context (alarm calls in particular in vertebrates).

Line 105-107: need references

=> The following reference to an exhaustive review has been added: Šobotník, J., Jirošová, A. & Hanus, R. Chemical warfare in termites. *J. Insect Physiol.* 56, 1012–1021 (2010).

Line 110: odd word choice “unforgettable”; unforgettable by who?

=> Everyone who has been bitten by a large termite soldier or stung by bullet ant know what this expression means. We prefer keeping this sentence as it is, however, we are disposed to tune it down upon direct request from the editor.

Line 110: “many personal observations”: be specific as whose observations you were referring to

=> The observations have been made mainly by DSD and JŠ in the field, this information has been added.

Line 107-114: would have been great to see the inclusion of some of the other amazing termite weapon types (e.g. nasutes, symmetrical and asymmetrical snapping mandibles etc.)

=> We added information about the nasus and the self-sacrifice of workers. However, to avoid to expand the text too much, only the existence of symmetrical and asymmetrical snapping mandibles has been mentioned. We are willing to expand more if requested.

Line 117: define “vibroacoustic communication”

=> Done.

Line 119: define “mechanical signaling”

=> This term is not important and appeared only once so we removed it.

Line 119-122: very long sentence. Consider splitting into 2 separate sentences

=> We split the sentence into two.

Line 123: remove “the” at start of sentence

=> Done.

Line 128: remove “the” before “vibroacoustic”

=> Done.

Line 130: remove “the” at start of sentence

=> Done.

Line 136: “direct observations”- vague. More detail needed

=> This term has been explained by adding that “In Nature, it is common to see individuals of some social insects warning their nestmates of potential danger.”

Line 137-140: long-winded sentence- consider splitting into 2 or rephrasing

=> We split the sentence into two.

Line 157: could include definitions for terminology used in the introduction here (e.g. vibroacoustic signalling, alarm signalling etc)

=> We used these terms before the terminology section, and so we defined such special terms at the first appearance in the text.

Results and Discussion

Figures could be improved by removing grey background (takes away from the main visual features of the figures).

=> The grey background has been removed from all figures and all tables. The figures are provided at low definition in the main text to ease the reading but they are also provided at high definition as separate files.

Colour scheme in Fig 4 should be changed as the black and blue colour of the boxplots can be hard to distinguish.

=> The color has been changed, we hope to better.

All figures also have 2 captions, so double check which one should be included in the final manuscript.

=> Done.

Fig 6 is missing an axis label on the x axis (vibroacoustic variables plot).

=> No, it is not. Both plots are the same including the labels, however, in the left one are emphasized the vibroacoustic characters (in blue) while in the right one the ecological ones (in green). We stressed this fact out in the figure heading.

As my experience with phylogenetics is limited, I am unable to comment on this aspect of the paper, however, it would be beneficial if you could include how you created the phylogenies used in Figure 2 and 3.

=> The phylograms are simplified from published phylogenies (refs 83, 109) as stated at line 534.

Line 178-179- remove “the” in front of general alarm

=> Done.

Line 185-187: not convinced by this statement “locomotion activity increased in many cases”- use explicit numbers here because from figure 1, it looks like changes in locomotion activity were not significant most of the time.

=> The locomotion activity increased in many cases but not most of the time, indeed. This increasing is clearly showed by the green arrows in figure 1 so we do not consider needed to add numbers to explicit the sentence. However, we added that this increasing is particularly obvious in most of the disturbances used in *Reticulitermes*.

Line 189-196: you define what the green fields and arrows mean, however you also need to define what the blue boxes with a x are to make things clear.

=> The blue fields with a cross indicate no significant difference. This information has been added.

Line 198-206: very interesting observation of termite defensive behaviours!

=> Thank you, we believe it too!

Figure 2: How was the tree in this figure produced?

=> General comparison of the evolutionary history of vibroacoustic and chemical alarm distribution among termites and the wood roach *Cryptocercus*. Topography based on published phylogenies (83, 109). Ancestral states counted according to Maximum Parsimony model. Dots are yellow when the alarm mode is present, white when absent, and grey when unknown. The family name colors represent ranks for MCA analysis.

Line 246: include “taxa” or “genera” after Neoisoptera

=> Word “genera” included.

Line 373-375: need references

=> The following references have been added:

Noirot CH (1970) The nests of termites. In: Krishna K, Weesner F, editors. Biology of Termites, New York and London: Academic Press, Vol 2. pp. 73–125.

Shellman-Reeve, J. S. The spectrum of eusociality in termites. in *The Evolution of Social Behaviour in Insects and Arachnids*, B. J. Crespi, J. C. Choe, Eds. (1997), pp. 52–93.

Conclusions/future directions

Whilst the authors have summed up the results of the study well, it would be nice to see more about the implications of this research as well as expanding on the future directions of this field as they are only briefly touched upon.

=> We added a few lines about the soldierless termites that have not been study for their alarm communication system but they are probably interesting since soldiers are lacking. In addition, we expanded the text to the comparison with alarm pheromones and vibrational communication in other eusocial groups.

The methods are somewhat lacking in detail and clarity, with important aspects missing from the section (i.e. how phylogenetic trees used in figures were constructed; sample sizes for behavioural experiments; how treatments were randomised etc.).

=> The sample sizes are included in Table S1, as mentioned in M&M section Behavioral experiments. Treatment randomization means that we have randomly selected 2 workers and 2 soldiers at the beginning of each experiment, i.e. 1 minute before the stimulus introduction, and these were analyzed for the subsequent movements throughout whole record. This information seems clear to us from the respective M&M section, but we are willing to expand the information upon the editor's request.

Line 446: include permit details

=> Done.

Line 454: sample sizes needed for experimental groups

=> This information is different according to the species so it was easier to include it in the Dataset S1 than in the text. To make the things clear, we added the following information in the main text: "for details on caste ratio, see Dataset S1".

Line 456-459: were treatments carried out by the same person each time? Variability in researcher technique when carrying out these treatments may influence results

=> Yes, all these experiments have been done by the same person, this information has been added.

Reviewer #2 (Remarks to the Author):

Among the 20 termite species studied, none includes the soldierless termite group. I think that will be interesting to include at least one Apicotermitinae species in discussion, since these termites have peculiar defensive strategies.

=> We added a few lines about soldierless termites in the discussion. It is indeed a missing group in our study, thank you for pointing it out.

In Material and Methods, it is important to be clear which species were studied in behavioral and vibroacoustic experiments.

=> This information is clearly presented in Table S1. We also added the following information in the material and methods section: The vibroacoustic experiments were carried out on all the species indicated in Figure 4. About behavioral experiments: Only the species whose caste ratio is indicated in Table S1 were studied.

Line 516 – 500,000 interactions (not “iterations”).

=> The proper term for chi-square test is iterations, i.e. the number of repetitions of the process.

I hope that my comments will be useful for the improvement of the manuscript.

=> All comments have been highly appreciated and the reviewers and the editor are acknowledged for their helpful and constructive comments at the end of the main text.

REVIEWERS' COMMENTS:

Reviewer #1 (Remarks to the Author):

The authors addressed all my concerns and the manuscript is now much easier to read. I don't have any other comments to add. I look forward to seeing it in print, it will be a great addition to the field of termite defence!